# Effect of Metal Ions on Hybrid Graphite-Diamond Nanowire Growth: Conductivity Measurements from a Single Nanowire Device

**DOI:** 10.3390/nano9030415

**Published:** 2019-03-11

**Authors:** Muthaiah Shellaiah, Ying-Chou Chen, Turibius Simon, Liang-Chen Li, Kien Wen Sun, Fu-Hsiang Ko

**Affiliations:** 1Department of Applied Chemistry, National Chiao Tung University, Hsinchu 300, Taiwan; muthaiah1981@nctu.edu.tw; 2Department of Electronics Engineering, National Chiao Tung University, Hsinchu 300, Taiwan; biolab@faculty.nctu.edu.tw; 3Department of Materials Science and Engineering, National Chiao Tung University, Hsinchu 300, Taiwan; turibius.icn@gmail.com (T.S.); fhko@mail.nctu.edu.tw (F.-H.K.); 4Center for Nano Science and Technology, National Chiao Tung University, Hsinchu 300, Taiwan; lcli@faculty.nctu.edu.tw

**Keywords:** nanodiamond, nanoparticles, metal ions induced graphitization, supramolecular assembly, graphite-diamond nanowires, carrier transport, carbon nanomaterials, conductivity, semiconductors

## Abstract

Novel Cd^2+^ ions mediated reproducible hybrid graphite-diamond nanowire (G-DNWs; Cd^2+^-**NDS1** NW) growth from 4-Amino-5-phenyl-4H-1,2,4-triazole-3-thiol (**S1)** functionalized diamond nanoparticles (**NDS1**) via supramolecular assembly is reported and demonstrated through TEM and AFM images. FTIR, EDX and XPS studies reveal the supramolecular coordination between functional units of **NDS1** and Cd^2+^ ions towards NWs growth. Investigations of XPS, XRD and Raman data show the covering of graphite sheath over DNWs. Moreover, HR-TEM studies on Cd^2+^-**NDS1** NW confirm the coexistence of less perfect sp^2^ graphite layer and sp^3^ diamond carbon along with impurity channels and flatten surface morphology. Possible mechanisms behind the G-DNWs growth are proposed and clarified. Subsequently, conductivity of the as-grown G-DNWs is determined through the fabrication of a single Cd^2+^-**NDS1** NW device, in which the G-DNW portion L2 demonstrates a better conductivity of 2.31 × 10^−4^ mS/cm. In addition, we investigate the temperature-dependent carrier transport mechanisms and the corresponding activation energy in details. Finally, comparisons in electrical resistivities with other carbon-based materials are made to validate the importance of our conductivity measurements.

## 1. Introduction

Research on nanowires (NWs) growth has become essential due to its diverse applications [1,2,3,4]. For example, conducting or semiconducting NWs have been effectively applied in photonics or solar cells [5,6,7]. Therefore, much focus has been given for the construction of metallic or semiconducting NWs [8,9,10]. Development in hybrid semiconducting NWs has also become essential for diverse electrochemical and photonic applications [11,12,13,14]. For instance, research on hybrid graphite-diamond nanowires (G-DNWs) in electrochemical, sensing and semiconducting utilities are still intensive [15,16,17,18]. Moreover, these G-DNWs are employed in many novel nanoelectronic devices [19,20,21]. However, the effectiveness of these nanodevices is attributed to the insertion of graphene sp^2^ carbon over the surface of diamond sp^3^ carbon, which acts as an insulator [22,23]. The conductivity of the graphite sheath enfolding over the insulator diamond in G-DNWs can be improved by tuning the sp^3^/sp^2^ carbon ratio [24]. But, these G-DNWs are generally grown by means of chemical vapor deposition (CVD) techniques [25,26,27,28], which require costly instruments to sustain high temperature. Therefore, it is essential to find alternatives for the fabrication of the G-DNWs in a cost-effective way.

As reported earlier, self-assembled diamond nanoparticles (DNPs) tend to form graphite sheath over NW-like structure via electrostatic interactions [29,30,31]. Moreover, graphitization over DNPs can be enhanced by laser, dangling bonds, photon energy, metal ions and pHs [32,33,34,35,36]. These diamond/graphite (sp^3^/Sp^2^) changes can be authenticated from Raman interrogations via in-plane bond stretching of sp^2^ graphite sheath represented by G-band coexisted with locally disordered forbidden D-band [37]. Moreover, doping of foreign substance, such as metal ions, may modify the sp^3^/sp^2^ ratios and lead to specific conductivity applications [38]. Recently, metal ion mediated self-assembly of DNPs that led to formation of hybrid G-DNW-like structure was reported [39,40]. In this way, ultrathin graphite-diamond nanorods with 2.1 nm diameter were fabricated from diamond clusters through self-assembly using microwave plasma chemical vapor deposition (MPCVD) technique and then applied in field emission studies [41]. In contrast, our group recently reported a cost-effective wet chemical route for the construction of G-DNWs via pH induced self-assembly of cysteamine functionalized DNPs (**ND-Cys**) [36]. In a similar fashion, a metal ion mediated self-assembly of **ND-Cys** DNPs are delivered to provide the G-DNWs-like structure [42] which allow us to continue the perfection of such G-DNWs.

To further improve the conducting nature of DNWs, the graphite sheath enfolding is enhanced by doping of S and N atoms over the surface. The presence of S and N atoms effectively increased the adsorption of metal ions with specific coordination as reported earlier by our group [42]. In this light, the presence of aromatic phenyl group was able to improve the conductivity via participation of its inner electronic conjugation. Hence, to expand the uniqueness and electrical conduction of formed G-DNWs, the DNPs were firstly functionalized by 4-Amino-5-phenyl-4H-1,2,4-triazole-3-thiol (**S1**, containing phenyl group, S and N atoms) via profitable wet chemical route to provide the **NDS1**. With the presence of phenyl unit, S and N atoms, metal ions are coordinated with certain extend to form the graphite-wrapped NW-like assemblies. However, only certain metal ions with the **NDS1** may form the G-DNWs with good reproducibility. For example, Cd^2+^ ions form the G-DNWs (Cd^2+^-**NDS1** NWs) at higher percentile with great reproducibility and smooth morphology than other ions. Similar to the fabrication of single Sb_2_Se_3_ NW [43], poly(3-hexylthiophene) NW [44] and G-DNW [36] towards thermoelectric or conductivity studies, a single Cd^2+^-**NDS1** NW device is successfully fabricated and the conductivity and electron transport properties are investigated in this work.

Herein, we report the Cd^2+^ induced self-assembly of novel 4-Amino-5-phenyl-4H-1,2,4-triazole-3-thiol (**S1)** functionalized diamond nanoparticles (**NDS1**) which lead to the G-DNWs (Cd^2+^-**NDS1** NWs) growth. The conductivity and electron transport mechanisms of the as-grown Cd^2+^-**NDS1** NWs are well demonstrated by a single Cd^2+^-**NDS1** NW device.

## 2. Materials and Methods 

### 2.1. General Information

Commercial nanodiamond powder (DNPs) with a size of 4–12 nm was purchased from BAOO-WEI INTERNATIONAL CO., LTD., Taichung, Taiwan. 4-Amino-5-phenyl-4H-1,2,4-triazole-3-thiol (**S1**) and Thionyl chloride (SOCl_2_) were obtained from Sigma Aldrich (Taiwan Merck Co., Ltd, Taipei, Taiwan). Anhydrous reactions were completed under nitrogen atmosphere by following the standard procedures. Appropriate drying agents were utilized in the distillation of solvents. Fourier transform Infrared spectroscopy (FTIR), Raman spectroscopy, scanning electron microscopy (SEM), transmission electron microscopy (TEM), zeta potential, energy dispersive spectrum (EDX) and X-ray photoelectron spectroscopy (XPS) were employed to identify and characterize the as-synthesized nanodiamond derivatives **(NDS1)**. SEM and EDX measurements were done by JEOL-JSM-6700 (JEOL Ltd., Tokyo, Japan). TEM and HR-TEM data were attained by JEOL-JEM-2100 and JEOL-JEM-2100F (JEOL Ltd., Tokyo, Japan), respectively. AFM data were taken in a tapping mode using a Digital Instrument model atomic force microscope (AFM D3100; EXW Charlotte, NC, USA). The size of **NDS1** NPs and its zeta potential were obtained from SEM, TEM and dynamic light scattering BECKMAN COULTER Delsa^TM^ Nano C particle analyzer (Beckman Coulter Inc., Brea, CA, USA). FTIR investigations were carried out by Perkin Elmer 100 FT-IR SPECTRUM ONE spectrometer (PerkinElmer Inc., Waltham, MA, USA). Raman interrogations were done with a combination of HOROBA (HORIBA FRANCE SAS, Palaiseau France), Lab RAM HR instrumental set up and a 488 nm solid state laser. The powder XRD data of p-ND, NDA and **NDS1** were achieved from BRUKER AXS D2 Phaser (a26-x1-A2BOE2B; Bruker AXS Inc., Madison, WI, USA). XPS measurements were performed by a Microlab-350 (Thermo Electron Corporation; Waltham, MA, USA). The synthetic procedures for **NDS1** and details on data collections of SEM, TEM, DLS, XPS, XRD, AFM, FTIR, Raman, calculations of reproducibility and stability of G-DNWs growth are provided in Appendix A. 

Following instruments were employed in the fabrication of single Cd^2+^-**NDS1** NW fabrication. The Oxford Instruments Plasmalab80Plus was employed for plasma-enhanced chemical vapor deposition of a 300 nm SiO_2_ layer over an N-type Si substrate. Next, double-side mask aligner (model: AG1000-4D-D-S-M-V; M&R Nano Technology co. Ltd, Taoyuan, Taiwan) was used in the electrode design. Subsequently, alignment marks and completion of electrodes interconnections were done using an electron beam lithography system (ELS-7500EX; ELIONIX INC., Tokyo, Japan). Dual-Beam focus ion beam microscope (FIB) (model: LYRA3; TESCAN, Brno – Kohoutovice, Czech Republic) was utilized to allocate the Cd^2+^-**NDS1** NW and to make electrical contact. The deposition of Ti/Au electrode contacts was done by a thermal evaporator TE-400. Finally, the electrical conductivity and temperature-dependent conductivity measurements from the metal-oxide-semiconductor field-effect transistor (MOSFET) structures were obtained from a cryogenic probe station (CG-196-Model; Everbeing Int’l Corp., Hsinchu, Taiwan) and a semiconductor device parameter analyzer (Keysight B1500; KEYSIGHT TECHNOLOGIES, Taipei, Taiwan).

### 2.2. Fabrication of a Single Cd^2+^-NDS1 NW

Allocation of a single Cd^2+^-**NDS1** NW was carried out by selecting a successfully grown nanowire via incubating 1 ng of **NDS1** in 1 mL of 1 nM of Cd^2+^ ions for approximately 45 min. During the allocating processes of a single Cd^2+^-**NDS1** NW, the cleaning process in each step was done by ultrasonication with Acetone, IPA and DI water followed by N_2_ purging to remove water. The steps for allocating a single Cd^2+^-**NDS1** NW are descripted as follows: (1) Selection of a heavily doped N-type silicon wafer as a substrate; (2) The selected Si-wafer was treated with H_2_SO_4_:H_2_O_2_ = 3:1 for 10 min to oxidize the organic materials on the surface and then washed by DI-water; The Si-wafer was further exposed to diluted HF to remove the oxide present on the surface; (3) Deposition of SiO_2_ (300 nm) on the Si-substrate by PECVD (model: plasmalab 80 plus; Oxford Instruments, Concord, MA, USA; at 300 °C, RF plasma: 25 W, Chamber pressure: 1000 mTorr, SiH 4:9 sccm, N_2_O: 710 sccm, reaction time: 250 s); (4) Completion of outside pad (Appendix A) via cleaning process, spin coating of hexamethyldisilazane (HMDS) and photoresist, UV lithography, thermal evaporation of 20 nm Ti and 60 nm Au, followed by lift off in acetone for more than 1 day. The completed 4-inch wafer was then cut into 2.2 cm pieces; (5) Completion of alignment mark (Appendix A) through cleaning process, spin coating the photoresist, E-beam lithography followed by thermal evaporation of Ti/Au (20/60 nm) and then lift off; (6) Deposition of dispersed Cd^2+^-**NDS1** NWs at 1 ng/mL of water; (7) SEM observation of the NW distributions; (8) Utilization of FIB to fix the single Cd^2+^-**NDS1** NW as well as making electrical contact; (9) Completion of electrode connections by means of cleaning process, spin coating the photoresist, E-beam lithography followed by thermal evaporation of Ti/Au (20/100 nm) and finally lift off. The above fabricated device was used in current-voltage (I-V), temperature-dependent conductivity and MOSFET measurements. 

### 2.3. Activation Energy (E_a_) Calculations

The activation energy [45] of diamond NW portion L2 was calculated by the following Equation:R = R_0_*exp*^(Ea/kT)^(1)
where R and R_0_ are the resistance of L2 with and without applied voltage, respectively, k is the Boltzmann constant and T is temperature in kelvin (K). By plotting the lnR as a function of 1/T (Appendix A), the activation energy (E_a_) can be determined from the slope of the curve.

### 2.4. Evaluation of Electron Transport Mechanisms

To determine the electron transport mechanisms [46] involved in the Cd^2+^-**NDS1** NW (L2 portion), conductivity and lnR were plotted as a function 1/T and T^-1/4^, respectively, where R stands for electrical resistance and T represents the temperature. Both curves were fitted linearly and nonlinearly. Similarly, lnσ was plotted as a function of T^−1/4^, which was then used to derive the relation of E as a function of T, where σ represents the conductivity (for our case obtained σ_0_ = 0.0077) and E stands for energy. The relation of E versus T can be fitted with Mott’s 3D hopping energy and thermal energy, which can be obtained from the following Equation: W_VRH_ = ¼ KT (T_M_/T)^1/d+1^(2)
where VRH stands for variable range hopping, kT is known as thermal energy, T represents temperature and T_M_ is described as Mott’s characteristic temperature (in this case T_M_ = 60603 K). The plot of lnσ *Vs* T^−1/4^ at temperature of 200–80 K can be fitted with three-dimensional (3D) Mott variable range hopping model (with d = 3 in Equation (2)).

## 3. Results and Discussions

As shown in Appendix A, slightly modified synthetic route [36,47] was followed to obtain **NDS1** from pristine nanodiamond (p-ND) particles. In which the synthesized ND-Acidchloride (NDA) was directly refluxed with excess 4-Amino-5-phenyl-4H-1,2,4-triazole-3-thiol (**S1**) to produce the **NDS1**. During synthesis, intermediates and **NDS1** were obtained with considerable yields. The equilibrium states of **S1** (Appendix A) may avoid the competitive reactions of acid chloride with sulfur group, which equilibrate as free thiol (-SH) or thione (=S). Hence, these ND-Acidchloride can easily react with free primary amine (-NH_2_) group to form the stable **NDS1** structure as shown in Figure 1. 

The as-synthesized **NDS1** was characterized by FTIR and Raman spectroscopy. As shown in Appendix A, –C=O stretching of **NDS1** is presented at 1695 cm^−1^, together with p-ND (at 1643 cm^−1^) and NDA (at 1601cm^−1^). Moreover, the stretching of –OH and –COOH from p-ND and NDA occur at 3413 and 3396 cm^−1^, respectively. However, the amide –NH from the **NDS1** is broadened from 3032 to 3400 cm^−1^. The aromatic –CH stretching of **NDS1** occurs at 2868 cm^−1^. Note that the free thiol (–SH) and amide –NH asymmetric bands of **NDS1** appear at 2562 and 1475 cm^−1^, respectively. On the other hand, free thiol (-SH) of **S1** is revealed at 2574 cm^−1^. The –NH bending stretch of **NDS1** appears at 1561 cm^−1^ in contrast to **S1** (-NH bending at 1529 cm^−1^). Other than the FTIR measurements, Raman interrogations also confirmed the formation of **NDS1**. As shown in Appendix A, D and G bands of p-ND and NDA appear at 1323 and 1605 cm^−1^, respectively. However, for **NDS1,** the D and G bands occur at 1374 and 1585 cm^−1^, respectively, wherein the intensity of the G band also demonstrates the occurrence of partial graphitization. The –COOH and amide –NH bands of NDA and **NDS1** are broadened from 3200 to 3600 cm^−1^. In a similar fashion, zeta potential of **NDS1** (Appendix A) further confirms the functionalization of **S1** over the nanodiamond surface. In contrast to the p-ND (−25.29 mV) and NDA (−27.94 mV), the **NDS1** possesses a positive zeta potential (+8.66 mV).

To further confirm the functionalization of **S1**, the elemental investigations were done by EDX and XPS. As illustrated in Appendix A and Table 1, the existence of S (5.47%) and N (17.71%) atoms with increased O (17.40%) and reduced C (59.42%) atomic concentrations proves that the **S1** is capped over ND surface. Likewise, the presence of these atoms is validated from the XPS of **NDS1** [Figure 2, Appendix A and Appendix A]. C1s peaks at 284.5 and 286.2 eV in the XPS spectra of C atom also imply the occurrence of partial graphitization. On the contrary, the manifestation N, O and S atoms are found by the N1s, O1s and S2p peaks at 401.5, 533.7 and 164.5 eV, respectively. Next, the morphology and particle sizes of the **NDS1** were investigated via SEM, TEM, AFM and DLS. As shown in Figure 3a, SEM image of **NDS1** reveals the diverse morphology than that of p-ND and NDA. Moreover, the particle size of **NDS1** at 100 µg/mL in water is 125.4 ± 76.9 nm as determined by dynamic light scattering (DLS) analysis (Appendix A). However, at much lower dispersed concentrations [48,49] (such as 1 ng/mL in water), DLS-based calculations become less effective. Therefore, TEM and AFM were used instead to determine particle sizes, as described in the following sections. 

TEM interrogation of **NDS1** at 1 ng/mL in water is displayed in Figure 3b, which confirms the existence of smaller size particles. As noted in Figure 3c,d, the AFM images of surface and height also confirm the small particles along with large particles. HR-TEM image shown in Figure 4a at 1 ng/mL in water confirms the presence of smaller **NDS1** agglomerated particles. Hence, we conclude that the particle size of **NDS1** must lies between 15~200 nm. The HR-TEM studies further prove the partial graphitization of nanodiamond occurred during functionalization of **S1** via the images of amorphous graphite along with the diffraction pattern of nanodiamonds. Figure 4b and the inset show the nanodiamond (111) diffraction distance of 0.206 nm and the amorphous graphite enfoldment [50]. In the HR-TEM analysis, the existence of defects or impurity channels may arise from the wet chemical synthesis of DNPs, which leads to the formation of less perfect sp^2^ graphite layer over DNWs. This issue will be discussed latter. The crystallinity of **NDS1** is authenticated by powder XRD analysis as displayed in Appendix A. The **NDS1** shows XRD peaks at 2θ = 43.59, 75.24 and 91.46 corresponding to the (111), (220) and (311) diffraction patterns of nanodiamonds, respectively. Moreover, a mild amorphous graphite peak can be seen at 2θ = 25.5, which is attributed to the partial graphitization of **NDS1** during the experiment.

The development of graphitized hybrid G-DNWs was done by incubating 100 µg of **NDS1** with 100 µM of cations (Na^+^, Ni^2+^, Fe^3+^, Cd^2+^, Ca^2+^, Ga^3+^, Cr^3+^, Cu^2+^, Fe^2+^, Mg^2+^, Au^3+^, Y^3+^, Al^3+^, Ag^+^, Co^2+^, Zn^2+^, Pb^2+^, Mn^2+^ and Hg^2+^) in 1 mL of water for 45 min as detailed in Appendix A. As shown in Appendix A, majority of the cations tend to form the NWs via metal ion mediated self-assembly of **NDS1**. However, in order to successfully conduct conductivity studies, highly reproducible NW (**NDS1** NW) formation is evaluated as discussed in Appendix A. From 100 collected data of G-DNWs with different metal ions, we concluded that **NDS1** NWs grown in the presence of Cd^2+^ ions showed 24% reproducibility (as shown in Figure 5), which was the highest among all ions. Above reproducibility can be further improved to 36% upon incubating the **NDS1** with Cd^2+^ ions for 24 h. However, further reproducibility enhancement beyond 36% cannot be achieved. As shown in the SEM images [Figure 6a,b and Appendix A], diverse Cd^2+^-**NDS1** NWs are formed at 10 µg of **NDS1** with 10 µM of Cd^2+^ ions per mL in water. On the other hand, at higher dispersive concentrations, such as 100 picogram (pg) of **NDS1** with 100 picomolar (pM) of Cd^2+^ ions, NW formation cannot be observed. Variations of either **NDS1** or Cd^2+^ have led to the agglomeration of particles.

The formed G-DNWs (Cd^2+^-**NDS1** NWs) were well characterized as described next. The width of Cd^2+^-**NDS1** NWs are between 50 to 980 nm with average lengths of 100 nm ~ hundreds of microns, depending on the dispersion concentrations which can be adjusted from 1 ng~100 µg of **NDS1** with 1 nM~100 µM of Cd^2+^ ions, correspondingly. Due to the multi-dispersive nature of **NDS1** with Cd^2+^ ions, longer G-DNWs are formed with larger diameters than those of earlier reports on diamond NWs with 5–20 nm in diameter [51]. However, with the variations in dispersion concentrations of **NDS1** from 100 µg to 1 ng and Cd^2+^ ions from 100 µM to 1 nM, respectively, the supramolecular self-assembled and scattered Cd^2+^-**NDS1** NWs with limited difference in lengths and diameters are formed, as shown in the TEM images [Figure 6c,d and Appendix A]. When the dispersion of **NDS1** and concentrations of Cd^2+^ ions were further diluted, the Cd^2+^-**NDS1** NWs formation can be affected due to lack of supramolecular interactions between them.

To establish the semiconducting applicability of Cd^2+^-**NDS1** NWs, characterization of these G-DNWs were carried out as described in the following. Firstly, FTIR spectral interrogations reveal supramolecular interactions between **NDS1** and Cd^2+^ ions. As shown in Appendix A, the –C=O stretching of **NDS1** is blue shifted to 1681 cm^−1^ and the amide –NH band is broadened between 3000 to 3600 cm^−1^. Note that the free –SH band of **NDS1** at 2562 cm^−1^ is almost vanished, which may be due to the coordinate bond between –SH of **NDS1** and Cd^2+^ ions. Moreover, the amide –NH bending and asymmetric stretching at 1561 and 1475 cm^−1^ are blue shifted to 1548 and 1459 cm^−1^, respectively. It indicates that the **NDS1** tends to form NWs in the presence of Cd^2+^ ions via feasible supramolecular interactions. The feasible repeating units in the Cd^2+^-**NDS1** NWs formation is schematically displayed in Appendix A. Due to the stability of the continual organometallic coordination [52] between **NDS1** and Cd^2+^ ions, the G-DNWs can be formed with exceptional morphology, greater lengths and diameters. However, such coordination becomes unstable at higher or lower concentrations or dispersion ratios which lead to mixed structures. This may be due to the irregular coordination of **NDS1** with Cd^2+^ ions at these dispersion ratios. Hence it confirms that, at fixed dispersion ratios, the G-DNWs growth with diverse diameters and lengths can be stabilized.

To explain the morphological and conductivity properties of formed G-DNWs, the appearance of Cd^2+^ ions over the surface of NWs is established from EDX, XPS and XRD spectral investigations. Preliminary analysis of EDX (Appendix A) and Table 1 demonstrate the slightly affected percentile of S (5.22%), N (14.46%), O (16.51%) and C (51.60%) than those of the originals, which proves the contributions of S, N and O atoms in the NWs formation. Wherein, reduced content of C atoms may be attributed to the improved graphitization of **NDS1** NWs. Moreover, alteration in the percentage of C, S, N and O atoms may be attributed to the dislocation of these atoms and the supramolecular coordination with Cd^2+^ ions, which generate few impurity channels or defects. The EDX spectrum also confirms the existence of Cd atom (12.21%), hence labelled as Cd^2+^-**NDS1** NWs. Above feasible supramolecular interactions might further accelerate the development of less perfect sp^2^ graphite layer over Cd^2+^-**NDS1** NWs which may improve the conductivity.

To demonstrate the improved graphitization, the manifestation of C, N, O, S and Cd atoms and the involved supramolecular coordination, XPS studies were conducted subsequently. As shown in Figure 7 and Appendix A, C1s peak in the XPS spectra of Cd^2+^-**NDS1** NWs is located at 284.3 eV in contrast to the original at 284.5 and 286.2 eV, which confirms the partial graphitization. Disappearance of the peak at 286.2 eV also implies the possible coordination of –C=O functional group with Cd^2+^ atoms. In a similar fashion, N1s, O1s and S2p peaks in the XPS spectra of Cd^2+^-**NDS1** NWs appear at different positions at 399.4, 532.2 and 163.2 eV (Appendix A) than that of **NDS1**, which also confirm their existence and involvement in supramolecular interactions. The contribution of Cd^2+^ ions and its participation towards supramolecular coordination in NWs formation is evident by the Cd3d peak at 405.2 eV in the XPS spectra displayed in Appendix A. Finally, as shown in Appendix A, the XPS interrogations support the proposed schematic from the FTIR analysis.

The possible supramolecular coordination is further established by TEM and AFM studies, subsequently. As shown in Appendix A, TEM images of **NDS1** with Cd^2+^ ions over carbon-copper grid reveal the self-assembly of **NDS1**, which show elongated NWs growth. Above images were obtained by scanning at different regions of C/Cu grid at 1 ng/mL dispersion of **NDS1** with 1 nM of Cd^2+^ ions. In a similar fashion, the AFM images (Appendix A) of Cd^2+^-**NDS1** NWs have also verified the self-assembly of **NDS1** at 1 µg/mL dispersion of **NDS1** with 1 µM of Cd^2+^ ions. These visualized self-assembly of **NDS1** is likely to improve the graphitization as explored in XPS analysis. Therefore, it is expected to find out enriched graphitization by means of XRD and Raman surveys. 

The improved graphitization during formation Cd^2+^-**NDS1** NWs was demonstrated through powder XRD investigation. Appendix A reveals the peaks at 2θ = 43.52, 75.28 and 91.16 degree corresponding to (111), (220) and (311) diffraction patterns of diamond, respectively, along with improved graphitization peak at 2θ = 26.29 degree in compared to the **NDS1**. These processes may also lead to the formation of impurity channels via coordinative binding of Cd^2+^ ions with **NDS1**, which may enhance the conductivity. Overall, results from EDX, XPS and XRD studies support the possible supramolecular interactions along with partial graphitization in the Cd^2+^-**NDS1** NWs growth. Next, to demonstrate the improved graphitization, we examined the NW-like assembly of **NDS1** mediated by all the metal ions using Raman spectroscopy as described in the following sections. 

For Raman studies, 100 µM of each metal ion was added to 100 µg/mL dispersion of **NDS1** and incubated for 45 min. The final spectra were obtained from 30 collected data via scanning at five different locations of six separate samples. In the presence of metal ions, Raman spectrum of **NDS1** exhibits the D and G bands between 1330~1360 and 1590~1610 cm^−1^, correspondingly, with diverse intensity as shown in Appendix A. Moreover, these cations also show the graphite band between 2600~3000 cm^−1^ as displayed in Appendix A. Based on the D and G band intensity, the graphitization ratios (I_G_/I_D_) are calculated and presented in Appendix A. From Raman investigations of **NDS1** with all metal ions, the Cd^2+^ ions show the improved graphitization via highly intense G band at 1585 cm^−1^ and less intense D band at 1381 cm^−1^. In contrast to other ions, Cd^2+^ ions reduce the D band intensity and enhance the G band intensity, which indicates the possible enhancement of graphitization. The intensity of the graphite band between 2600~3000 cm^−1^ of **NDS1** with Cd^2+^ ions is clearly stronger than those of other ions. As presented in Appendix A, the I_G_/I_D_ ratio of Cd^2+^-**NDS1** NWs is 1.27, which is greater than those **NDS1** NWs assemblies formed with the remaining metal ions. Hence it is chosen for the following electrical conductivity measurements. Note that the described graphitization enrichment in Cd^2+^-**NDS1** NWs, which plays vital role in the conductivity studies, may also affect its morphology along with the formation of impurity channels. The surface morphology and impurity channels were visualized by HR-TEM measurements at 1 ng/mL dispersion of **NDS1** with 1 nM of Cd^2+^ ions as explained in the following.

The graphite sheath wrapping over the diamond core of Cd^2+^-**NDS1** NWs and the existence of impurity channels are well demonstrated by HR-TEM and FT diffraction analysis, as shown in Figure 8. HR-TEM images of a single Cd^2+^-**NDS1** NW show improvement in morphology, which may due to a rigid diamond core. Under high magnifications, diffraction distances of 0.206 nm and 0.126 nm are determined and are assigned to (111) and (202) pattern of diamond, respectively. Moreover, above HR-TEM images (Figure 8) also show impurity channels or defective voids (green circles) along with graphite sheath (red circles). These observations suggest that the formed NWs possess more diamond-like characteristics with slight enrichment of graphite and impurity channels, which lead to longer NWs formation in contrast to previous reports. These impurity channels may arise from the presence of Cd^2+^ ions and wet chemical synthesis. The visualization of diamond core of Cd^2+^-**NDS1** NWs through the HR-TEM studies is not considerably affected by the presence of amorphous and less perfect sp^2^ graphitic layer and impurity channels, which also propose fewer graphite sheath wrapping. On the contrary, the G-DNWs grown by the CVD may have less perfect sp^2^ graphite layer and impurity channels over sp^3^ diamond core at diverse ratios. Nevertheless, the Cd^2+^-**NDS1** NWs with more diamond-like characteristics must have lower conductivity and will be discussed later. 

From FTIR, XPS, TEM and AFM studies, the mechanism behind the Cd^2+^-**NDS1** NWs growth can be explained as follows. In the presence of Cd^2+^ ions, the DNWs growth is initiated through supramolecular interactions with the functional units (–C=O, –NH and –SH) of partially graphitized **NDS1** particles, which may further induce the self-assembly and lead to formation of NWs. During the Cd^2+^-**NDS1** NWs growth, Cd^2+^ ions act as initiative and supramolecular binding source to boost the self-assembly and graphene shells formation and impurity channels. These graphene shells and impurity channels may appear over the surface and further sandwich between the **NDS1** particles and hence generate less perfect sp^2^ graphite wrapping over the DNWs. The presence of these graphene shells on the less perfect graphite layer and the impurity channels might act as the connecting units between diamond cores of the DNWs. Due to the supramolecular self-assembly of dissimilar sized **NDS1**, the Cd^2+^-**NDS1** NWs are grown with various lengths and diameters. However, the supramolecular assembly may enhance rigidity by avoiding the formation of excess graphene shells and impurity channels over the NWs, which reveal more diamond-like characteristics. Due to the rigidness, smoothness and diamond-like characteristics, the conductivity of Cd^2+^-**NDS1** NWs may be affected considerably as demonstrated next.

To authenticate the hybrid G-DNWs formation, conductivity measurements were conducted on a single Cd^2+^-**NDS1** NW device (as shown in Figure 9A) on a Si/SiO_2_ wafer via the fabrication processes illustrated in Figure 9B. The fabrication processes include pads design, alignment marks, DNWs dispersion, Au/Ti contacts deposition and interconnections of single NW (Appendix A) as previously described in Section 2.2. SEM image of the selected single Cd^2+^-**NDS1** NW with designed structures of N-Si/SiO_2_/Cd^2+^-**NDS1** NW/Au is displayed in Figure 10a. The selected Cd^2+^-**NDS1** NW has an overall length of 7.7 µm with a diameter of 400 nm. The single Cd^2+^-**NDS1** NW is divided into three sections, namely, L1 (2.22 µm), L2 (1.12 µm) and L3 (1.34 µm) by the contacts 1 to 4.

A 196-Model cryogenic probe station was utilized to evaluate the conductivity properties of L1, L2 and L3. The resistivity and conductivity from the I-V characteristics were calculated [53] using following Equations (3) and (4).
σ = 1/ρ(3)
and
ρ = R (A/l)(4)
where σ represents the conductivity in S/cm and ρ is recognized as the static resistivity obtained from Equation (4); R is known as electrical resistance of a specimen in Ω, A represents the cross-sectional area of the specimen in m^2^ and l is the length of material in meter.

These NW portions were scanned between −4 and 4 V and their conductivity was measured in air (760 torr) and vacuum (10^−2^ torr) conditions, as shown in Figure 10b,c. As displayed in Table 2, the measured conductivity of L2 in air and vacuum possesses a conductivity of 1.15 × 10^−4^ and 2.31 × 10^−4^ mS/cm with coefficient of variations of 20% and 11%, respectively. These values are better than those of L1 and L3 under the same conditions. As shown in Figure 10d,e, the obtained conductivity results further confirm that L2 shows a better conductivity in vacuum than that of in air. The dissimilar behavior in conductivity of the L1 and L3 portions is evident due to the irregular graphite sheath wrapping. Different degree of graphite sheath/graphene shells coverage over L1 and L3 is attributed to the supramolecular assembly of the **NDS1** NPs with dissimilar sizes in the presence of Cd^2+^ ions, as illustrated in aforementioned HR-TEM studies. However, the conductivity obtained from L2 is lower than that of pH induced G-DNWs [36] because of the utilization of diverse functional units and growth approaches. Wherein, the **NDS1** with altered functional units are supramolecularly assembled in the presence of Cd^2+^ ions and induce NWs growth with more diamond-like characteristics and mild graphite sheath wrapping. The more diamond-like characteristics may lead to the growth of rigid NWs with smooth morphology as evident in the HR-TEM images but with lower conductivity in compared to the **ND-Cys** NWs. To investigate the transport mechanisms in L2, temperature-dependent conductivity (I-V) measurements were engaged between 80 ~ 300 K. The conductivity decreases with decreasing temperature as shown in Figure 11 and Appendix A.

The observed linear and symmetrical I-V behaviors of L2 throughout the entire temperature range indicate good ohmic contacts. The obtained conductivity is resulted from the sp^2^ graphene shells, impurity channels and Cd^2+^ ions over the G-DNW surface and in the core. The conductivity may also be affected by the presence of sulfur (S) and Nitrogen (N) atoms on the surface of Cd^2+^-**NDS1** NWs. The temperature-dependent electrical resistance and static resistivity decrease as the temperature is increased from 80 K to 300 K, as shown in Appendix A. By fitting the plots of “lnR vs. 1/T” (Appendix A), the activation energies (E_a_) with values of 29.41 meV and 12.43 meV for L2 are determined for temperature ranges of 300~200 K and 200~80 K, respectively, which clearly indicates that two kinds of transport mechanisms are involved [54]. In addition, the plot of “lnσ *Vs* 1/T” (Appendix A) also support the manifestation of different transport mechanisms in the temperature-dependent conductivity studies. In order to further investigate the electron transport mechanisms, plots of “conductivity *Vs* 1/T” and “lnR *Vs* T^-(1/4)^” are fitted nonlinearly and linearly, as shown in Appendix A. Similarly, a plot of “lnσ *Vs* T^-(1/4)^” (Figure 12a) is converted to a plot of “energy (E) *Vs* temperature (T)” (Figure 12b). By using the aforementioned Equation (2), it can be confirmed that, between 200 ~ 80 K, the electron transport is dominated by 3D-mott variable-range hopping [55]. However, at temperature above 200 K, the transport mechanism is governed by thermal activation [56].

Between 200 ~ 80 K, hopping of electrons is supported by the coexisting impurity channels/Cd^2+^ ions and graphene shells over **NDS1** surface. On the other hand, between 200~300 K, the electrical conductivity is mostly contributed by the thermally activated impurities or less perfect sp^2^ graphite layer over Cd^2+^-**NDS1** NW. Finally, the transconductance measurements between −4 to 4 V from the metal–oxide–semiconductor field-effect transistor (MOSFET) [57] structures of L1, L2 and L3 reveal that the NW conductivity cannot be modulated by the applied gate bias (Appendix A) due to the inferior quality of the interface and/or poor carrier concentrations of the Cd^2+^-**NDS1** NWs. 

The drain-source current (I_ds_) is completely dominated by the gate leakage current when gate bias is higher than 15 V because of the poor NW/SiO2 interface. However, modulation of the transconductance of the MOSFET will be possible if suitable dielectric material with improved interface properties with the NWs was found in the future. As summarized in Table 3, values of our NW resistivity are in between those of the diamond carbon nanotubes and diamond based electrical resistivity studies [58,59,60,61,62,63,64]. The current Cd^2+^-**NDS1** NWs also have a lower resistivity than that of our previous reported uneven G-DNW (**ND-Cys** NW) via pH-induced assembly. These Cd^2+^-**NDS1** NWs possess rigidness and smooth surface morphology, which lead to the insufficient sp^2^ graphitic layer over diamond and hence slightly reduced conductivity. Therefore, we are currently working towards the development of novel G-DNWs by combining both tactics that can cultivate G-DNWs with both flatten morphology and improved electrical conductivity. 

## 4. Conclusions

The 4-Amino-5-phenyl-4H-1,2,4-triazole-3-thiol (**S1**) functionalized diamond nanoparticles (**NDS1)** is synthesized and employed in metal ions mediated self-assembly directed hybrid graphite-DNWs growth. In the presence of Cd^2+^ ions, G-DNWs are formed with 24% reproducibility, which is higher than that of other metal ions. The Cd^2+^-**NDS1** NWs have diameters between 50 to 980 nm and lengths between 100 nm ~ hundreds of microns. Involvement of supramolecular coordination in the self-assembly of Cd^2+^ contributed **NDS1** particles towards NWs growth is supported by FTIR, EDX and XPS results. Graphite wrapping, impurity channels and existence of Cd^2+^ ions over the DNWs surface are clearly demonstrated by EDX, XPS, Raman, XRD and TEM results. The Cd^2+^ mediated supramolecular assembly of **NDS1** towards G-DNWs construction is well supported by TEM and AFM investigations. The as-grown G-DNWs show rigid diamond characteristics with smooth morphology, which is evident in the HR-TEM images. A single Cd^2+^-**NDS1** NW device is fabricated to conduct I-V measurements and to determine its applicability. In vacuum, the conductivity of L2 section in the single G-DNW is determined with a value of 2.31 × 10^−4^ mS/cm. Temperature-dependent conductivity studies of the single Cd^2+^-**NDS1** NW portion L2 reveal that the carrier transport mechanism is dominated by 3D-mott variable-range hopping between 200~80K with an activation energy of 29.41 meV. However, at higher temperature between 200~300 K, the transport mechanism becomes thermally activate with an activation energy of 12.43 meV. This new strategy for the growth of rigid G-DNWs with smooth morphology may further promote the diamond NW-based semiconductor research.

## Figures and Tables

**Figure 1 nanomaterials-09-00415-f001:**
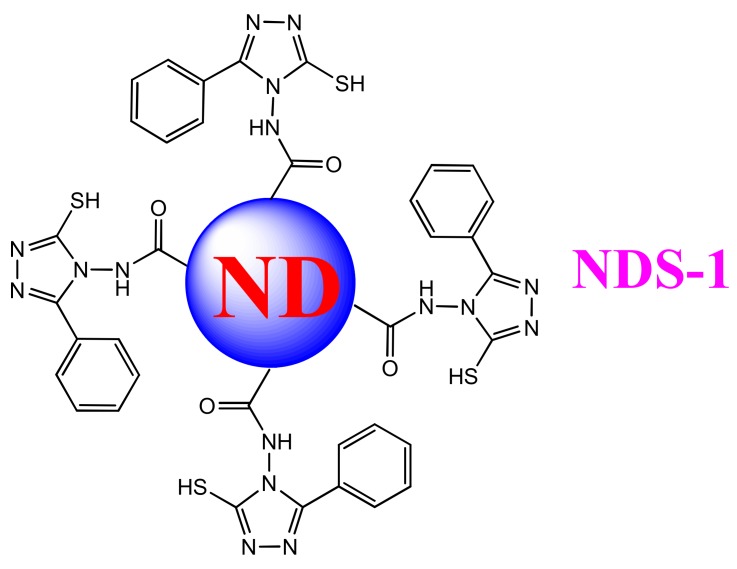
Model structure of **NDS1** obtained from pristine nanodiamond (p-ND) particles.

**Figure 2 nanomaterials-09-00415-f002:**
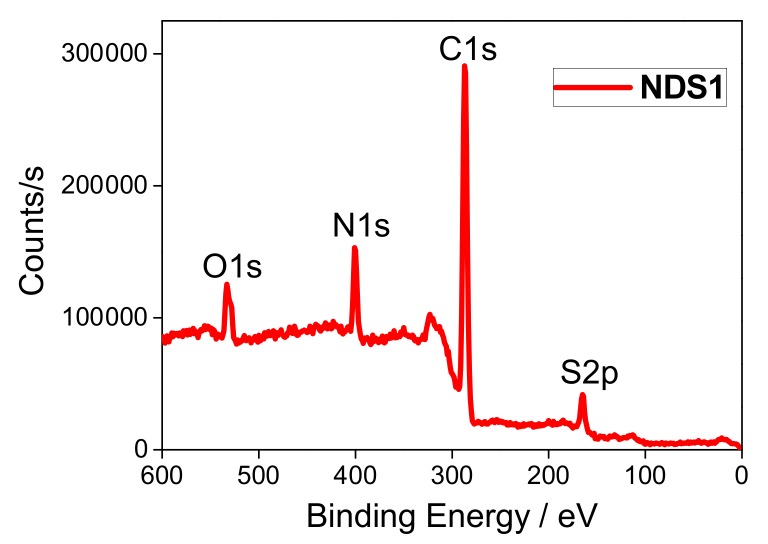
XPS spectrum of **NDS1** confirms the presence of Carbon (C1s), Nitrogen (N1s), Oxygen (O1s) and Sulfur (S2p) atoms.

**Figure 3 nanomaterials-09-00415-f003:**
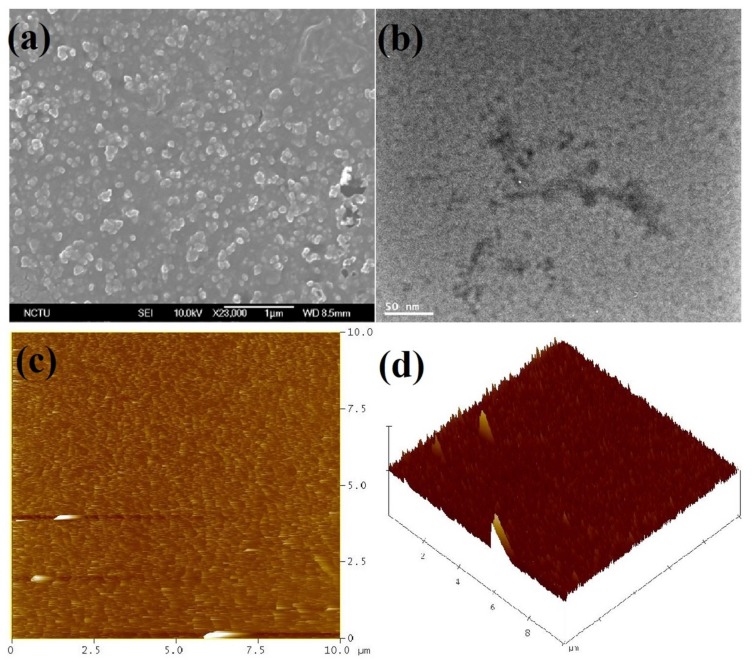
(**a**) Scanning electron microscopy (SEM) image of **NDS1** at 100 µg/mL dispersion (scale: 1 µm); (**b**) Transmission electron microscopy (TEM) image of **NDS1** at 1 ng/mL dispersion (scale: 50 nm); (**c**) Atomic force microscopy (AFM) image of **NDS1** at 10 µg/mL dispersion (scan range: 0–10 µm) and (**d**) AFM height image of **NDS1** at 10 µg/mL dispersion (scan range: 0–10 µm).

**Figure 4 nanomaterials-09-00415-f004:**
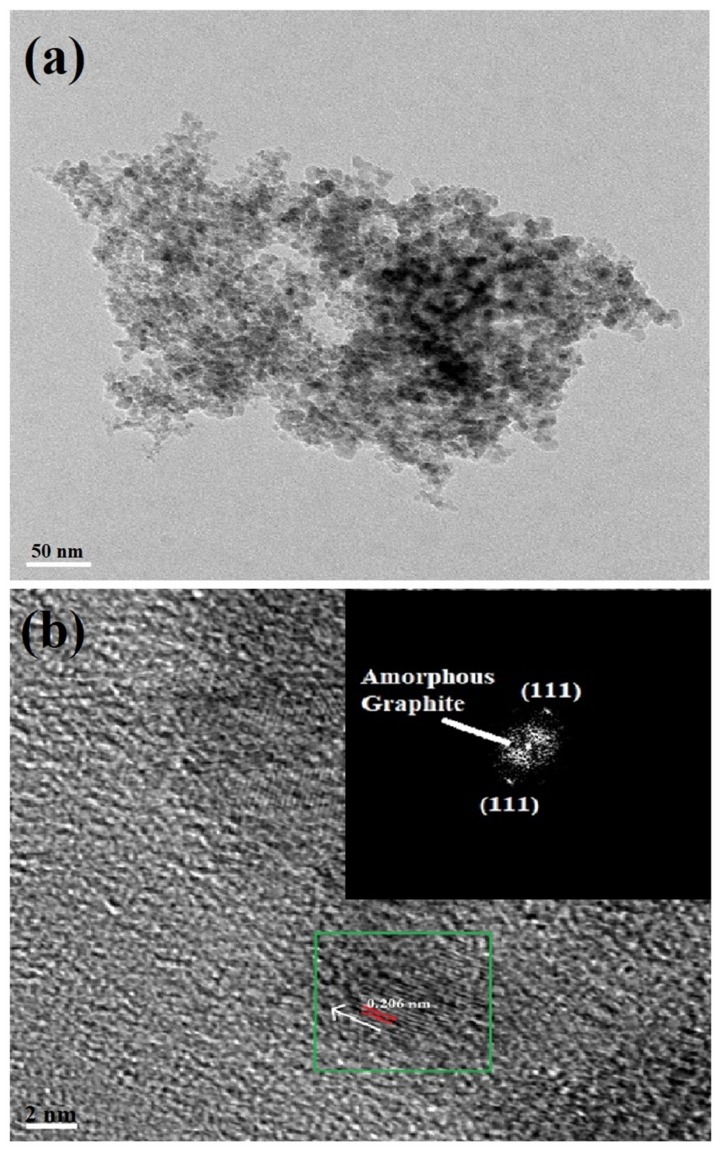
(**a**) HR-TEM image of **NDS1** at agglomerated state shows possible small particles and (**b**) Diffraction distance 0.206 nm represents the (111) pattern of nanodiamond along with amorphous graphite existence.

**Figure 5 nanomaterials-09-00415-f005:**
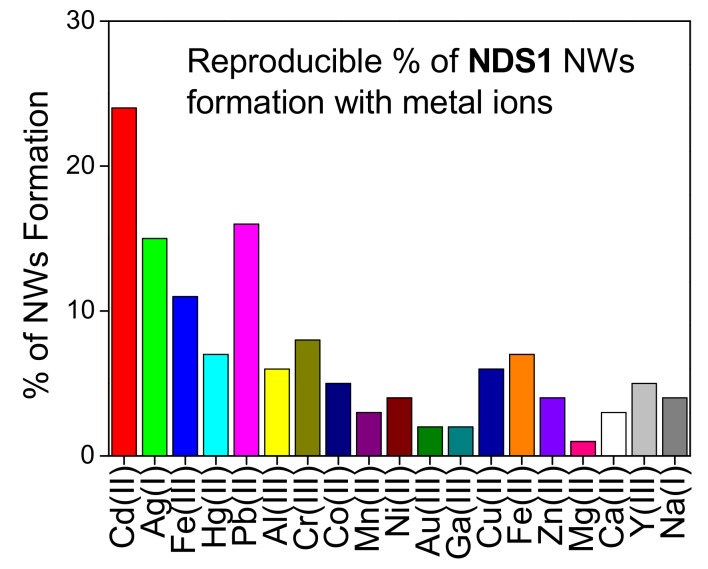
Reproducibility percentage from 100 collected data on DNWs growth with respect to different metal ions.

**Figure 6 nanomaterials-09-00415-f006:**
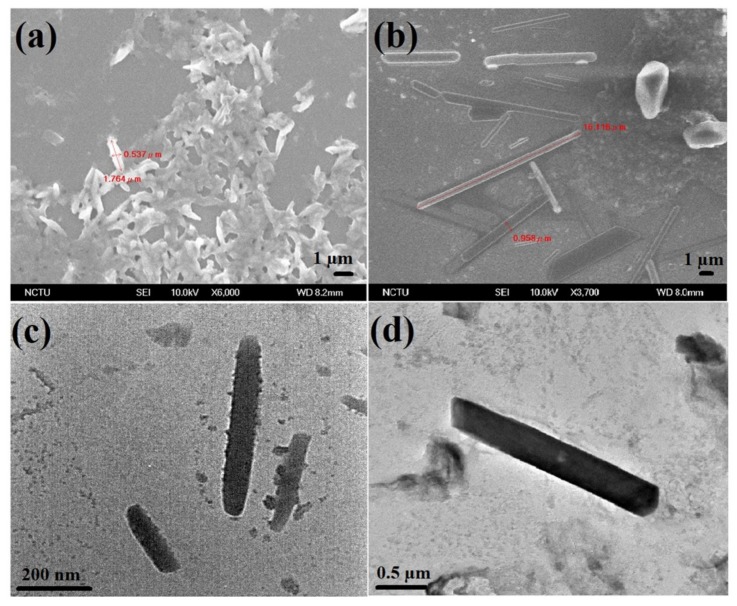
(**a**) and (**b**) SEM images of Cd^2+^-**NDS1** NWs (10 µg/mL in water) at different regions (scale: 1 µm) (**c**) and (**d**) TEM images of Cd^2+^-**NDS1** NWs (1 ng/mL in water) at different regions (scale: 200 nm and 0.5 µm, correspondingly).

**Figure 7 nanomaterials-09-00415-f007:**
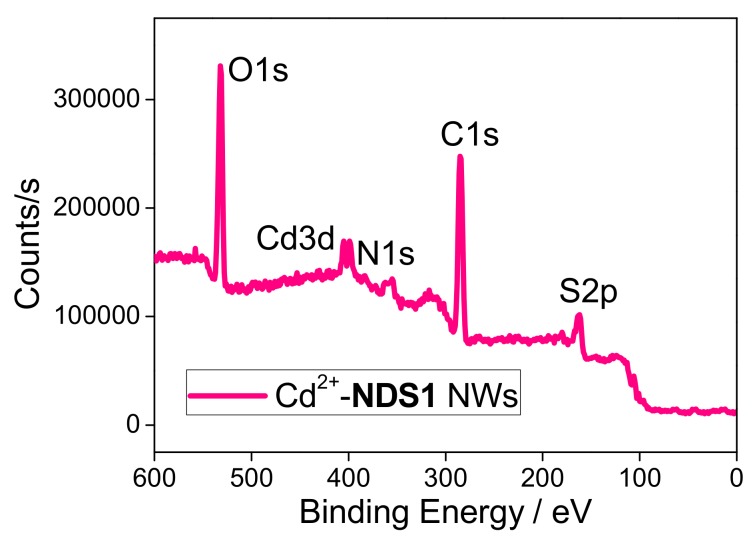
XPS spectrum of Cd^2+^-**NDS1** NWs demonstrates the presence of Carbon (C1s), Nitrogen (N1s), Oxygen (O1s), Sulfur (S2p) and Cadmium (Cd3d) atoms.

**Figure 8 nanomaterials-09-00415-f008:**
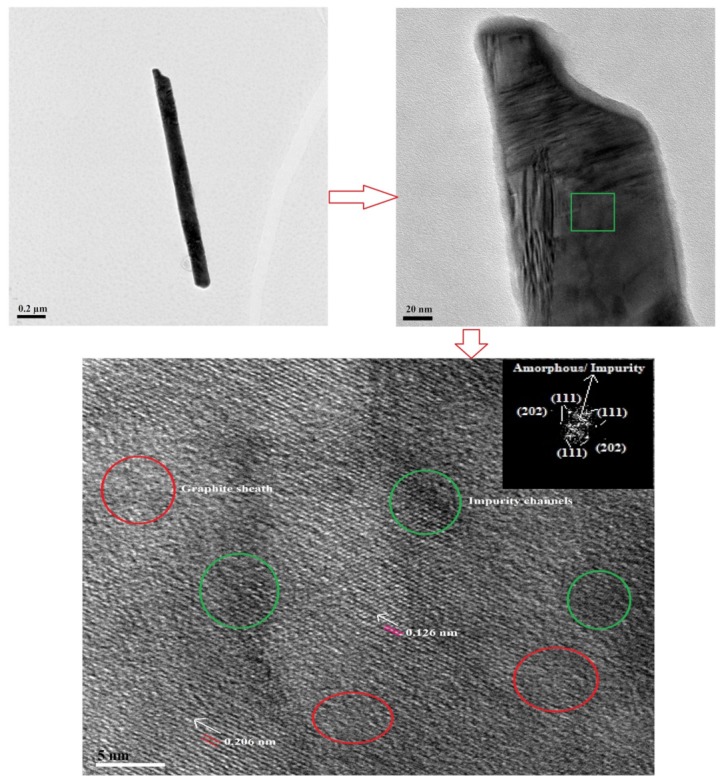
HRTEM images of single Cd^2+^-**NDS1** NW obtained at 1 ng/mL dispersion in water indicate the presence of graphite sheath (red circles) and impurity channels (green circles). The diffraction distances 0.206 nm and 0.126 nm are related to (111) and (202) patterns of nanodiamond (Scale: 0.2 µm, 20 nm and 5 nm, correspondingly).

**Figure 9 nanomaterials-09-00415-f009:**
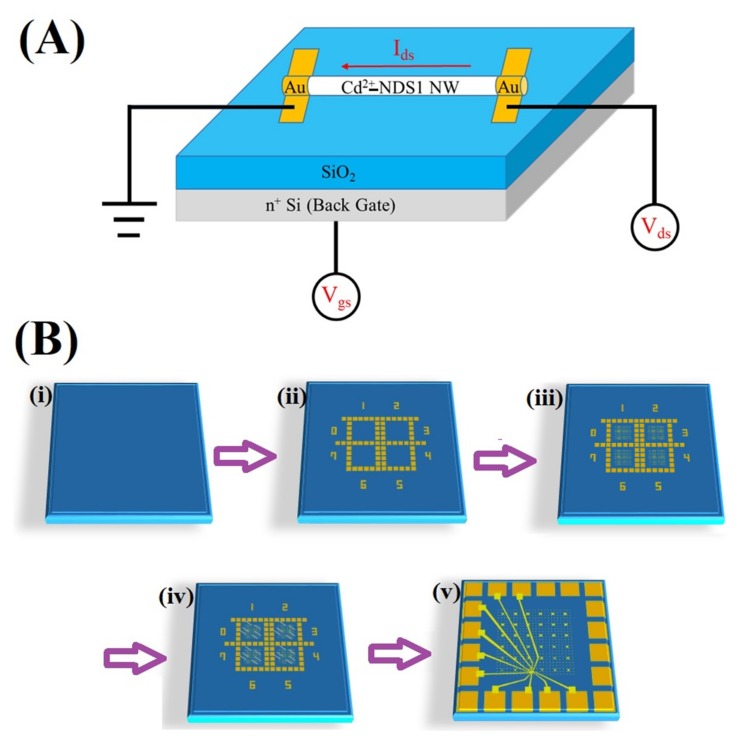
(**A**) Schematic of a single Cd^2+^-**NDS1** NW device and (**B**) Steps involved in fabrication are (i) PECVD 300nm SiO_2_ on N-type Si (Resistance:0.02Ω), (ii) Pad patterning by NUV lithography, (iii) Alignment mark patterning by E-beam lithography (iv) Nanowires dispersion followed by FIB and (v) Interconnection patterning by E-beam lithography.

**Figure 10 nanomaterials-09-00415-f010:**
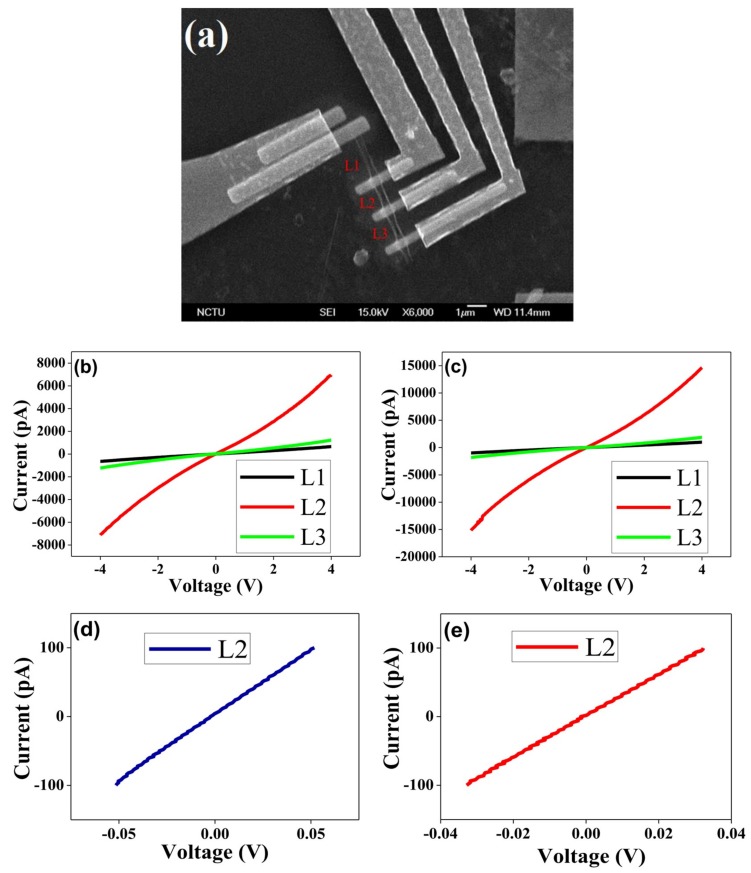
(**a**) SEM image of the single Cd^2+^-**NDS1** NW with Au contacts. I–V curves of L1, L2 and L3 measured using a two-point probe at 300 K in (**b**) air (760 torr) and (**c**) vacuum (10^−2^ torr). I–V curves of L2 measured using a four-point probe at 300 k in (**d**) air (760 torr) and (**e**) vacuum (10^−2^ torr).

**Figure 11 nanomaterials-09-00415-f011:**
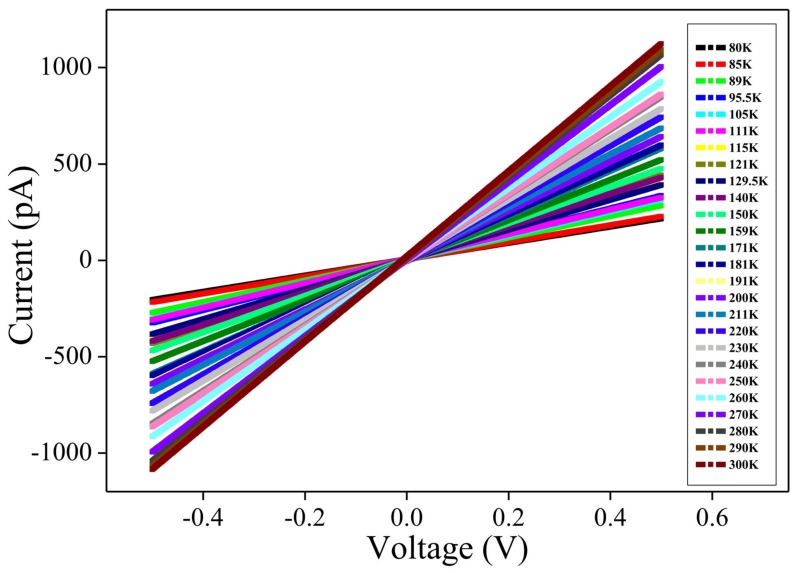
Temperature-dependent current-voltage (I–V) curves for Cd^2+^-**NDS1** NW-L2 (2-contacts, 2-point probe in vacuum (10^−2^ torr)) between 80 ~ 300 k.

**Figure 12 nanomaterials-09-00415-f012:**
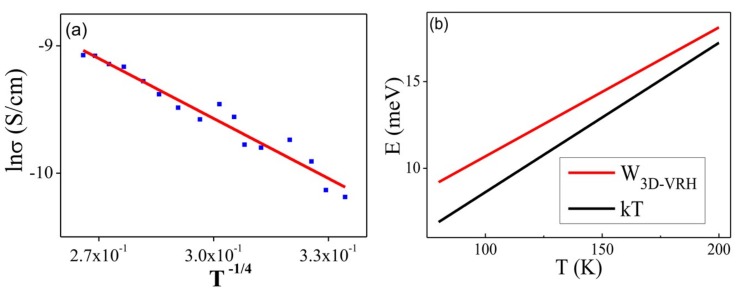
(**a**) Plot of “lnσ *Vs* T^−1/4^” for Cd^2+^-**NDS1** NW (L2) between 200–80 K and (**b**) Plot of “E *Vs* T” derived from “lnσ *Vs* T^−1/4^” represents the linear fittings of “Mott’s 3D hopping energy (red line)” and thermal energy (black line) obtained from Equation (2).

**Table 1 nanomaterials-09-00415-t001:** EDX data of ND derivatives and Cd^2+^-**NDS1** NWs.

Compound	C (%)	O (%)	N (%)	S (%)	Cd (%)
p-ND	96.21	3.79	-	-	-
NDA	87.32	12.68	-	-	-
**NDS1**	59.42	17.40	17.71	5.47	-
Cd^2+^-**NDS1** NWs	51.60	16.51	14.46	5.22	12.21

**Table 2 nanomaterials-09-00415-t002:** Resistance, resistivity and conductivity data of L1, L2 and L3 sections of a selected Cd^2+^- **NDS1** NW with four contacts.

Device Measurement Condition	Cd^2+^-NDS1 NW Area	Resistance (G·Ω)	Resistivity (Ω·cm)	Conductivity (mS/cm)	Coefficient of Variation
Atmosphere (760 torr)	L1	7.02	3.98 × 10^4^	2.51 × 10^−5^	0.66%
	L2	0.78	8.70 × 10^3^	1.15 × 10^−4^	0.20%
	L3	4.39	4.10 × 10^4^	2.44 × 10^−5^	0.51%
Vacuum (10^−2^ torr)	L1	4.55	2.58 × 10^4^	3.88 × 10^−5^	0.74%
	L2	0.39	4.33 × 10^3^	2.31 × 10^−4^	0.11%
	L3	2.91	2.72 × 10^4^	3.68 × 10^−5^	0.58%

**Table 3 nanomaterials-09-00415-t003:** Comparative account on electrical resistivities of Cd^2+^- **NDS1** NW with respect to selected diamond materials, carbon nanotube and carbon nanowire from I-V measurements.

Materials	Growth Route/Technique	Electrical Resistivity (Ω-cm)	Ref
Single **ND-Cys** G-DNW	Wet-Chemical Synthesis	370.76	[36]
**ND-Cys** + Hg^2+^ assembly	Wet-Chemical Synthesis	NA	[42]
Undoped-UNCD	NA	10^6^	[58]
Single Crystal Diamond	Commercial Source	10^14^	[59]
Ultra-thin Nanocrystalline Diamond Films	MPCVD	5 × 10^13^	[60]
Nitrogen Incorporated Diamond Films	MPCVD	10^5^	[61]
Single Diamond Nanowire	APCVD	NA	[62]
Carbon Nanotube	Carbon-arc Method	10^−3^~10^−6^	[63]
Carbon Nanowire	NA	0.015	[64]
Single Cd^2+^-**NDS1** NW	Wet-Chemical Synthesis	4.33 × 10^3^	This Work

NA = Not Available.

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
