# Peer review of "Effect of Metal Ions on Hybrid Graphite-Diamond Nanowire Growth: Conductivity Measurements from a Single Nanowire Device"

_nanomaterials, 2019, doi:10.3390/nano9030415_

Reviewer 1 Report

The paper reports the Cd2+ induced self-assembly of functionalized diamond nanoparticles, which are used to grow hybrid graphite-diamond nanowires (G-DNWs). XPS, XRD, Raman, AFM and HR-TEM analysis are performed to convincingly show that a graphite sheath covers the diamond nanowire. Moreover, the conductivity and electron transport mechanisms of the so-produced graphite-diamond nanowires are investigated using a single-nanowire device.

The study can contribute to the development of hybrid semiconducting nanowires, which are currently an intense subject of research. The paper contains many interesting experimental data, generally well interpreted. However, the manuscript results too long, tedious and slightly confusing. The paper could be suitable for the publication after a major revision.

The authors should clearly identify the key messages and organize the paper accordingly. Too many inessential details and several repetitions should be avoided. In my opinion, both the experimental and discussion sections can be shortened.

An example: Lines 425-430 (“During the fabrication process, pads and alignment marks were completed as described in AutoCAD diagrams (Figures S20 and S21 in supporting information), followed by DNWs dispersion as shown in Figure S22a in supporting information. Next, four Au/Ti contacts were deposited over the selected single Cd2+-NDS1 NW as displayed in Figure S22b in supporting information. Finally, interconnections were done as prescribed in AutoCAD diagram (Figure S23a in 429 supporting information) and the interconnected single NW is shown in Figure S23b in supporting 430 information.”) can be omitted without loss of information since the fabrication of the device has been already described in section 2.9.

The same for Lines 437-440 (“Where σ represents the conductivity in S/cm and ρ is recognized as static 437 resistivity, which can be obtained from

ρ = R (A/l) ---------------- (4)

Where ‘R’ known as electrical resistance of a specimen in Ω, ‘A’ represents the cross-sectional area of the specimen in m2 and ‘l’ is the length of material in meter.”) being both formulas obvious.

The discussion and the investigation of the conduction mechanisms seem to be an important part of the study and should be better organized and improved. Some key plots, supporting the proposed conduction mechanisms should be included in the paper. Conversely, useless picture s such as 11 a or 12 b can be omitted, as 12a can be used as an inset (are the Fig 12a data referred to the two or four-probe configuration? Are they in air or atmosphere?).

The measurement conditions and the difference between the measurements in atmosphere and in vacuum (which pressure?) should be better discussed.

Furthermore, the authors conclude that the current of the graphene-diamond nanowire cannot be modulated by the gate voltage. However, they sweep the gate only between 0 and 10 V, that is not enough in my opinion to draw such a conclusion. 300 nm SiO2 can sustain up to 70-80 V. The authors could try a -70V to + 70 C gate sweep.

Author Response

Reviewer#1

Remarks to the Author 

The paper reports the Cd2+induced self-assembly of functionalized diamond nanoparticles, which are used to grow hybrid graphite-diamond nanowires (G-DNWs). XPS, XRD, Raman, AFM and HR-TEM analysis are performed to convincingly show that a graphite sheath covers the diamond nanowire. Moreover, the conductivity and electron transport mechanisms of the so-produced graphite-diamond nanowires are investigated using a single-nanowire device. The study can contribute to the development of hybrid semiconducting nanowires, which are currently an intense subject of research. The paper contains many interesting experimental data, generally well interpreted. However, the manuscript results too long, tedious and slightly confusing. The paper could be suitable for the publication after a major revision.
We are very grateful to the reviewer, for the appreciation and providing the opportunity to clarify our results, which improvise the manuscript further. 

The authors should clearly identify the key messages and organize the paper accordingly. Too many inessential details and several repetitions should be avoided. In my opinion, both the experimental and discussion sections can be shortened. 

An example: Lines 425-430 (“During the fabrication process, pads and alignment marks were completed as described in AutoCAD diagrams (Figures S20 and S21 in supporting information), followed by DNWs dispersion as shown in Figure S22a in supporting information. Next, four Au/Ti contacts were deposited over the selected single Cd2+-NDS1 NW as displayed in Figure S22b in supporting information. Finally, interconnections were done as prescribed in AutoCAD diagram (Figure S23a in 429 supporting information) and the interconnected single NW is shown in Figure S23b in supporting 430 information.”) can be omitted without loss of information since the fabrication of the device has been already described in section 2.9.

The same for Lines 437-440 (“Where σ represents the conductivity in S/cm and ρ is recognized as static 437 resistivity, which can be obtained from

ρ = R (A/l) ---------------- (4)

Where ‘R’ known as electrical resistance of a specimen in Ω, ‘A’ represents the cross-sectional area of the specimen in m2 and ‘l’ is the length of material in meter.”) being both formulas obvious.

 “Author Reply”

As suggested by the reviewer, we have provided the essential details and deleted unnecessary points. We have shifted the synthetic and some experimental procedures (previously sections 2.2 - 2.8) and Figure 8 to supporting information and shorten the manuscript length. Further, we have also modified those lines (Lines 425-430 and 437-440) without loss of information.

The discussion and the investigation of the conduction mechanisms seem to be an important part of the study and should be better organized and improved. Some key plots, supporting the proposed conduction mechanisms should be included in the paper. Conversely, useless picture s such as 11 a or 12 b can be omitted, as 12a can be used as an inset (are the Fig 12a data referred to the two or four-probe configuration? Are they in air or atmosphere?). 

“Author Reply”

As per the reviewer’s recommendation, discussion on conduction mechanisms well organized and improved in the revised format. Figure 11a in previous manuscript has been omitted and Figure 12b is shifted to supporting information as Figure S25. Figure 12a modified to Figure 11 and detail of probe configuration, contacts and pressure [2-contacts, 2-point probe in vaccum (10-2torr)]has been delivered in the caption. Plot of “lnσ Vs T-(1/4)” (Figure 12a) and “energy (E) Vs temperature (T)” (Figure 12b) is currently included, which supports the conduction mechanisms.

The measurement conditions and the difference between the measurements in atmosphere and in vacuum (which pressure?) should be better discussed. 

“Author Reply”

The measurement pressure condition of atmosphere and in vacuum has been clarified and discussed in text, Figures caption and Table 2. 

Furthermore, the authors conclude that the current of the graphene-diamond nanowire cannot be modulated by the gate voltage. However, they sweep the gate only between 0 and 10 V, that is not enough in my opinion to draw such a conclusion. 300 nm SiO2 can sustain up to 70-80 V. The authors could try a -70V to + 70V gate sweep.

“Author Reply”

As mentioned by reviewer, attempts has been maddened previously to scan between -70V to + 70V. But,“due to the poor NW/SiO2interface, the drain-source current (Ids) is completely dominated by the gate leakage current when gate bias is higher than 15 V”.

Reviewer 2 Report

the group from Taiwan did interesting work. Manuscript can be accepted only if the authors consider below remarks:

figure captions are too simple. Authors should give more info in figure captions.

EDX data should be given as a plot.

figure 6 sem & tem scales are not readable. (also fig 9)

figure 12 a some colours of temperatures are doubled..how we will distinguish?

fig 12 x axis temeprature unit should be (K) not (k)

some sp3/sp2 discussion should be given from literature in intro see and  cite: Physical Chemistry Chemical Physics 11 (27), 5628-5633 (2009); Journal of Applied Physics 117 (15), 153905 (2015)

Author Response

Reviewer#2

Technical Comments to the Author

The group from Taiwan did interesting work. Manuscript can be accepted only if the authors consider below remarks:

We are grateful to the reviewer, for providing the valuable comments to improve the technical standard of our manuscript. 

Figure captions are too simple. Authors should give more info in figure captions.

 “Author Reply”

As recommended by the reviewer, Figure captions are modified with some details.

EDX data should be given as a plot.

 “Author Reply”

EDX data with plot has been provided in Figures S7 and S16 in supporting information. 

Figure 6 sem & tem scales are not readable. (also fig 9)

 “Author Reply”

The Scale bars of those Figures are supplemented in readable form along with intimation in captions.

Figure 12a, some colours of temperatures are doubled, how we will distinguish?

 “Author Reply”

The colours of all temperatures are delivered in a distinguishable format in the revised manuscript and Figure 12a modified as Figure 11.

Figure 12b axis temeprature unit should be (K) not (k)

 “Author Reply”

As mentioned by the reviewer,the unit has been modified to (K) and the figure shifted to supporting information as Figure S25.

 Some sp3/sp2 discussion should be given from literature in intro see and cite: Physical Chemistry Chemical Physics 11 (27), 5628-5633 (2009); Journal of Applied Physics 117 (15), 153905 (2015)

 “Author Reply”

As suggested, the discussion on sp3/sp2 is provided as follows with those references [37] and [38].

“These diamond/graphite (sp3/Sp2) changes can be authenticated from Raman interrogations via in-plane bond stretching of sp2graphite sheath represented by G-band coexisted with locally disordered forbidden D-band [37]. Further, doping of foreign substrates such as metal ions may tune sp3/Sp2ratio and led to specific conductivity applications [38]”.

Round  2

Reviewer 1 Report

The authors have considered the reviewer's suggestions and improved the paper accordingly.

The article can be accepted for the publication.

Reviewer 2 Report

The manuscript can be accepted.